# Transition to Distance Learning: Student Experience and Communication during the COVID-19 Pandemic in the United Arab Emirates

Soumaya Abdellatif [1], Aizhan Shomotova [2,*], Safouane Trabelsi [3], Salwa Husain [4], Najeh Alsalhi [5] and Mohamed Eltahir [5]

1 Department of Sociology, College of Humanities and Sciences (CHS), Humanities and Social Sciences Research Center (HSSRC), Ajman University, Ajman P.O. Box 346, United Arab Emirates
2 Foundations of Education Department, College of Education, UAE University, Al Ain P.O. Box 15551, United Arab Emirates
3 Department of Sociology, College of Human and Social Sciences of Tunis, University of Tunis, Tunis 1938, Tunisia
4 College of Interdisciplinary Studies, Zayed University, Abu Dhabi P.O. Box 144534, United Arab Emirates
5 Research and Graduate Studies, Department of Education, College of Humanities and Sciences, Humanities and Social Sciences Research Center (HSSRC), Ajman University, Ajman P.O. Box 346, United Arab Emirates
* Correspondence: aizhanshom@gmail.com or 201990089@uaeu.ac.ae

**Abstract:** The COVID-19 pandemic in 2020 prompted higher education institutions in the United Arab Emirates (UAE) to switch to online learning for the safety of their citizens. The main purpose of this study is to determine the relationship between four indicators of digital learning experience and the intensity of student socio-pedagogical communication after the transition to distance learning. The data were collected from Ajman University, a private university in the UAE, during the spring of 2020. The sample consisted of 381 students who were surveyed using an online survey tool or email. First, this study found that the majority of students had access to digital tools and the Internet; however, a small number struggled with weak and unreliable Internet connection. Most students had a moderate to high ability to use digital technology, but some encountered difficulties and required assistance. Most students utilised digital communication tools for over five hours daily. The study also found a general lack of digital competency among students and difficulties in using digital tools for remote learning, highlighting the importance of investing in the development of digital skills. The study also found an intensification of social relationships and an increase in communication frequency between students and instructors; however, inadequate instructor–student communication remained a challenge. Finally, the multiple linear regression model showed that indicators such as the communication dimension of the lessons and the participatory nature of the courses positively impacted the intensity of student communication after the transition to distance learning.

**Keywords:** distance learning; online learning; university; higher education; students; digital; communication; United Arab Emirates; Ajman University; COVID-19; pandemic

## 1. Introduction

The COVID-19 pandemic led to a widespread shift toward online learning globally, including in the United Arab Emirates (UAE). With a population of 10 million in 2022, the UAE is located in Southeast Asia and Eastern Arabia [1] and is one of the wealthiest and most prosperous Arab nations, having rapidly developed since the discovery of oil in the 1960s. To compete globally and establish a knowledge-based economy, the UAE places a strong emphasis on providing high-quality education and developing innovative programmes in higher education [2,3].

Before the COVID-19 epidemic, higher education (HE) institutions in the UAE had made substantial investments in e-learning technologies and curricular integration [4,5].

E-learning is the dissemination of instructional content using digital means. Colleges and universities faced the challenge of incorporating online learning into their curriculums to equip their students with the skills needed for a tech-driven future. Although a lot of resources have been invested into edtech, advancements in e-learning have been slow and insubstantial [4]. However, only a few UAE higher education institutions have adopted entirely online courses or hybrid e-learning models [4,6,7]. Despite this, prior research indicates that students prefer face-to-face learning over blended learning methods [8,9]. Previous studies on the challenges and opportunities of online learning in the context of the COVID-19 pandemic recommended more research to examine the impact of the pandemic on the higher education system [8,10].

This study focuses on the shift to online learning during the COVID-19 pandemic at Ajman University, a private university founded in 1988 in the UAE. Before the COVID-19 pandemic, Ajman University used a combination of conventional face-to-face and video-conference learning modes, as well as the learning management system (LMS) Moodle [9]. Similar to other universities in the UAE, Ajman University followed the policies of the Ministry of Education and enforced 100% distance learning to prevent the spread of the pandemic within the university. One week before the shift, faculty members underwent training and simulated online courses to prepare for complete remote online teaching [11], while students received advice and instructions on the use of technology and tools for online learning.

A significant number of previous studies have examined online learning during the pandemic in higher education (HE) in the UAE [11,12]. However, the worldwide change from conventional learning to online education demands further study and an in-depth knowledge of the viewpoints of the institution's many stakeholders. Specifically, student perspectives on their learning experiences are essential for the continuous improvement of the HE system. Therefore, this paper first describes the experience and challenges of students during the transition to distance learning, including the readiness of the students' digital infrastructure, the ability of the students to use digital tools and their frequency of use, distance communication channels, the previous experience of students on online learning platforms, the distant examination and level of understanding of the courses, and the impact of the transition to distance learning in the event of an emergency at Ajman University in the UAE during the COVID-19 pandemic. Finally, the study presents the main findings on the relationship between four indicators of digital learning experience and the intensity of student socio-pedagogical communication between students, their classmates, and professors after the transition to distance learning.

This study begins with a comprehensive review of the literature, providing a global perspective on the transition to distance and online learning experience and challenges. It then reviews the studies conducted in higher education institutions (HEIs) in the UAE. Finally, the study presents the detailed methodology, followed by two subsections containing the results, discussion, and conclusion. This study provides important insights for academic institutions, policymakers, and educators on the importance of ensuring higher quality education through various modalities and of providing positive learning experiences for students.

## 2. Literature Review

### 2.1. University Student Experience and Challenges with Distance and Online Learning: Global Perspective

Previous studies have highlighted that the most significant challenges during the transition to remote learning were the availability and quality of the Internet, the availability of appropriate software, the quality of the e-learning system, hardware and other resources, and the digital proficiency of both educators and learners [11–13]. HEIs with a long history of online teaching and learning tend to receive positive evaluation from university students regarding online courses compared to those with less of a tradition in online learning [14]. Similarly, a study during the COVID-19 pandemic showed that e-learning aided in the

learning process of students; however, it placed a greater burden on them [15]. Therefore, student digital information and ICT skills may play a role in their adaptation to online learning. The results of a study that examined the relationship between student digital competence and academic engagement during the COVID-19 crisis and whether informal digital learning plays a mediating role in this relationship indicated that there was a positive and significant correlation between student digital competence and academic engagement [16]. In addition, informal digital learning played a mediating role in the relationship between students' digital competence and academic engagement.

More importantly, university students reported a lack of engagement and communication with instructors and classmates, difficulties in seeking help through online channels, and distractions in their learning environments [17]. However, some students reported that online learning benefited from the flexibility to learn at any place and time, saved time, and facilitated easy communication with instructors and students (such as breakout room discussion and private Zoom chat) [17]. Furthermore, one of the primary concerns raised by students at different universities in EU countries during online learning was socialisation issues, including a lack of flexibility and inadequate feedback from instructors [14]. Therefore, the attractiveness of online courses depended greatly on engaging and stimulating materials, and the interactive content of the courses was an important factor in how students assessed the effectiveness of their instructors. In synchronous online learning, students felt significantly more connected to each other, experienced a stronger sense of instructor presence, and scored higher on self-evaluation performance [18]. Furthermore, the degree of social presence in synchronous online courses was a significant predictor of student self-evaluation, academic performance, and students' sense of belonging to the university. On the contrary, in asynchronous learning mode, social presence had a weaker effect on predicting student learning outcomes.

### 2.2. University Students' Experience and Challenges with Distance and Online Learning in the UAE

The literature review showed a good number of studies that examined the transition to online learning during the pandemic in the UAE context. University students experienced higher-than-average information quality and system quality when moving to remote learning [19]. Additionally, students who were familiar with using technology for educational purposes viewed e-learning as helpful during the COVID-19 pandemic. They recognised its usefulness in times of crisis and when dealing with personal and external pressures during the pandemic. The research showed that students who were satisfied with the e-learning system had a desire to continue using it post-pandemic, especially those who were working. This indicates a positive attitude towards the e-learning method implemented during the COVID-19 pandemic.

Another study explored the perspectives of students at Al Ain University toward remote teaching and learning [5]. The study found that the participants had attitudes toward remote learning tools and technology. The university had already adopted these tools and online exams for some courses prior to the COVID-19 pandemic, allowing a smoother transition to distance learning during the pandemic. Students were satisfied with this new mode of learning as they viewed it as an opportunity to improve their ICT skills and the quality of their education. They also noted benefits such as COVID-19 safety, flexibility, and access to recorded sessions. However, the study also highlighted some challenges faced by students with online learning, such as unequal access to technology, a focus on improving ICT skills instead of subject matter, technical and internet connectivity issues, lack of hands-on experiences, and reduced interaction with instructors. In addition, students in the humanities and social sciences had more positive perceptions of online learning compared to those in scientific fields such as engineering and pharmacy. Overall, the research showed that students were generally satisfied with the university's readiness for distance learning.

Likewise, a study examined the effect of e-learning on student academic achievement at the University of Science and Technology of Fujairah in the UAE [20]. It was found that students' academic success was positively related to e-learning, e-learning adoption, and instructors' attitudes towards e-learning. The study also showed that student engagement mediates the relationship between student attitudes and academic achievement, as well as the relationship between student digital readiness and academic achievement. This suggests that incorporating engaging content into online learning can influence student engagement and success.

In particular, first-year students had limited experience with a blended learning system, while fourth-year students were more familiar with it at the College of Fine Arts and Design in the UAE [21]. Despite this, both groups of students still found full-time distance learning to be a new experience. Both groups reported high levels of stress and anxiety while working on design projects during the quarantine period.

Another study examined the attitudes of Al Ain University students towards remote learning during the early stages of the COVID-19 pandemic [22]. The students cited time and cost efficiency, safety, flexibility, and increased participation as the biggest advantages of online learning. However, they also reported distractions, a heavy workload, technical issues, and a lack of support from instructors and peers as the main drawbacks. The study highlights that the COVID-19 pandemic gave students and educators their first experience with online learning, emphasising the importance of investing in preparation and quality e-learning to ensure positive educational outcomes in the event of future disruptions.

Previous studies on remote education in the UAE have shown a lack of research on student perspectives on online learning during crises. This study adds to the existing literature on distance and online learning during emergencies in the UAE and the results can help educators provide a better digital infrastructure and a distant learning experience for future emergency situations. Additionally, the results can inform curricular enhancements and instructional adaptability during unexpected events that impact students' academic experiences.

## 3. Methods

### 3.1. Data Collection

Data were collected using an online survey tool in the spring of the 2020–2021 academic year by emailing all students from Ajman University in the UAE. There were 6346 students at Ajman University enrolled in that semester. As a result of the random sampling method, the final sample was n = 381 students.

### 3.2. Research Questionnaire

A questionnaire has been developed based on previous studies and included three sections. The first section consisted of sociodemographic questions. The second section consisted of five themes: 1—assessment of student digital infrastructure readiness, 2—assessment of students' ability to handle digital tools and their frequency of use, teaching methods, and channels for distance study, 3—evaluation of students' experience with distance learning platforms, 4—reflection of emergency distance learning experience on results, and 5—the level of course understanding. All questions were closed-ended with multiple-choice answers. The third part consisted of a single matrix of items rated on the Likert scale from 1 (strongly disagree) to 4 (strongly agree). In total, there were nine elements that measured the level of access to digital infrastructure and remote learning platforms, the frequency of use of digital tools and the Internet, the experience of communication, interaction, and discussion with classmates and teachers/professors, and the impact of moving to a distance learning system on academic outcomes during the transition to distance learning under the COVID-19 pandemic outbreak and lockdown (see Appendix A, Table A1).

### 3.3. Data Analysis

Development and Validation of the Instrument

The questionnaire was developed in Arabic as it is the main language of instruction for the population being studied, i.e., Arab-speaking students. It was reviewed by three faculty experts from the sociology department and a statistician to improve its content. A pilot questionnaire was administered to the first 30 students; however, these data were not included in the final sample. The validity of internal consistency was calculated based on the data collected and the reliability was tested.

The internal consistency of the questionnaire was assessed by computing the Pearson correlation coefficient between each item and the total score of the corresponding factors. The results showed that three indicators, the first, fourth, and fifth, were excluded because their correlation values were below 0.5.

The reliability of the questionnaire was measured by internal consistency, with results showing a minimum of 0.678 and a maximum of 0.812 (Table 1). The overall internal validity was 0.803, indicating a high level of reliability. Based on these results, the modified questionnaire was considered to have sufficient validity for use on the entire sample.

**Table 1.** Items that were extracted from the instrument.

| Indicator | Variables | Pearson r | *p*-Value | Alpha Coefficient | Number of Items |
|---|---|---|---|---|---|
| 2 | Frequency of use of digital infrastructure | 0.516 | 0.01 | 0.678 | 2 |
| 3 | The level of employability of digital infrastructure | 0.570 | 0.01 | 0.726 | 2 |
| 6 | The impact of moving to a distance learning system on academic outcomes | 0.684 | 0.01 | 0.812 | 2 |
| 7 | The communication dimension of lessons | 0.625 | 0.01 | 0.769 | 2 |
| 8 | Interactive dimension of lessons (centrality of discussion) | 0.559 | 0.01 | 0.714 | 2 |
| 9 | Intensity of student socio-pedagogical communication after the transition | 0.638 | 0.01 | 0.778 | 2 |
| | **Total Cronbach Alpha** | **0.803** | | | **18** |

### 3.4. Multiple Regression Analysis Model

The dependent variable, the intensity of student socio-pedagogical communication after the transition to distance learning, is measured using indicator 9. The four independent variables being studied are: 1—frequency of use of digital technology and internet surfing (indicator 2), 2—the impact of the transition to distance learning on academic outcomes such as level of course understanding and exam results (indicator 6), 3—the communication aspect of lessons (continuity of communication) (indicator 7), and 4—the interactive aspect of lessons (focus of discussion) (indicator 8). Multiple linear regression analysis was used to determine whether there are statistically significant relationships between the independent variables and the dependent variable. The study has four hypotheses, as shown in Figure 1.

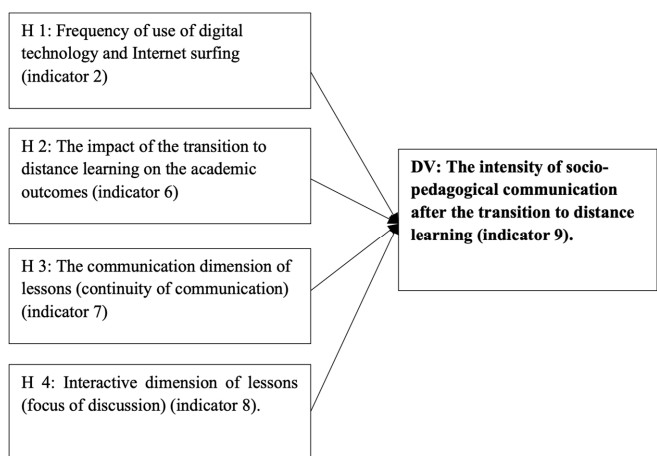

**Figure 1.** The conceptual framework of the study.

## 4. Results

### 4.1. Sample Demographic Data

In the study, 381 students participated and responded to the online questionnaire. The sample consisted of 63% female and 37% male students, with the majority of the respondents being between 21 and 25 years old (45.4%) followed by those between 16 and 20 years old (29.7%). The participants were from different universities, most of them from the Faculty of Humanities and Science (28.9%), followed by the Faculty of Engineering and Information Technology (19.2%) and the Business School (12.1%). Most of the students (54.9%) were in the second level of the undergraduate programme, with the rest in the first academic level (38.6%). Most of the students lived in urban areas, with 96% of the total sample residing in such areas.

### 4.2. Assessment of the Readiness of the Students' Digital Infrastructure

During the transition to online learning due to the pandemic, most of the students had access to a variety of digital technologies such as cellphones, laptops, desktops, and desktop computers. All the students at Ajman University in the UAE had personal laptops, and 81.4% of them also owned smartphones (Table 2). Over 79% of the participants in the study had multiple digital communication tools.

**Table 2.** Level of access to digital communication tools.

| Type of Digital Communication Tool | Yes | |
|---|---|---|
| | **N** | **%** |
| Smartphone | 310 | 81.4 |
| Numeric pad | 92 | 0.1 |
| A fixed computer for personal use | 76 | 19.9 |
| A laptop computer for personal use | 381 | 100 |
| A stationary computer for shared use | 72 | 18.9 |
| More than one tool | 302 | 79.3 |

Furthermore, the study found that less than half of the sample (42.6%) had a strong and stable Internet connection during the shift to remote learning. However, 34.2% of the students reported having a strong but unstable Internet connection quality. Therefore, in general, more than 75% of the respondents had good access to the Internet during distance learning, while only 23% reported a weak connection.

### 4.3. Assessment of Students' Ability to Use Digital Tools and Their Frequency of Use

The ability to utilise digital communication tools was rated on a three-point scale, from high ability to difficult to use. On average, 34.4% of the participants had a good level of skill with digital technology (Table 3). For instance, 52.8% reported a high proficiency with digital communication tools, while 12.9% reported a low level of proficiency. The average score was 1.6 with a standard deviation of 0.706, indicating a generally high level of competence with digital equipment.

Furthermore, this research reveals the frequency with which digital communication technologies are used while browsing the Internet. In general, 52.5% of the respondents have access to a high-speed Internet connection. However, 37% show a decent internet connection and only 10.5% of the simple have a poor one. In this context, the arithmetic mean is 1.58, and the standard deviation is 0.674, indicating a high degree of internet access in general. The indicator data reveal an overall arithmetic mean of 1.58 and a standard deviation of 0.61. This demonstrates the great capacity of the digital infrastructure offered to students at the AU.



**Table 3.** The level to use ability of digital technology and the level of Internet connection available during the transition to distance learning.

| | | Indicator 3: The Level of Digital Infrastructure Capacity | | | | | | |
|---|---|---|---|---|---|---|---|---|
| **Items** | **Indicator** | **High Level** | **Moderate Level** | **Low Level** | **Mean** | **SD** | **Order** | **Interpretation** |
| Level of ability to use digital devices | n | 201 | 131 | 49 | 1.60 | 0.706 | 1 | High ability |
| | % | 52.8 | 34.4 | 12.9 | | | | |
| Level of access to the Internet connection | n | 200 | 141 | 40 | 1.58 | 0.674 | 2 | High access |
| | % | 52.5 | 37.0 | 10.5 | | | | |
| **Overall mean** | | | | | 1.59 | 0.612 | | High-level |

The second indicator refers to the extent of using digital technology and the Internet. The overall results indicate a high frequency of use, with an average score of 1.72 and a standard deviation of 0.698 (Table 4). This suggests that students use the Internet and digital devices on a daily basis, enhancing their skills and ability to interact with technology regardless of the context.

**Table 4.** Frequency of use of digital technology and Internet surfing.

| | | Indicator 2. Frequency of Use of Digital Technology and Internet Surfing | | | | | | |
|---|---|---|---|---|---|---|---|---|
| **Items** | **Indicator** | **High Level** | **Moderate Level** | **Low Level** | **Mean** | **SD** | **Order** | **Interpretation** |
| Frequency of use of digital tools | n | 217 | 100 | 64 | 1.60 | 0.760 | 2 | High frequency |
| | % | 57.0 | 26.2 | 16.8 | | | | |
| Frequency of Internet surfing | n | 172 | 100 | 109 | 1.83 | 0.844 | 1 | Moderate frequency |
| | % | 45.1 | 26.2 | 26.2 | | | | |
| **Overall mean** | | | | | 1.72 | 0.698 | | High frequency |

### 4.4. Channels of Distance Communication

During the shift to remote learning, various communication methods were accessible due to the advancement of technology and digitalization. However, within the context of formal education, students primarily used email to communicate with their professors and classmates. This study found that the majority of students (58.5%) communicated with their professors through university emails while a small percentage used both university and personal emails (Table 5). On the other hand, 90% of the students used the WhatsApp application to communicate with their classmates during distance learning, and 34.7% even used it to communicate with their professors. These results show that students mainly use official channels, such as email, for communication with their professors and prefer informal channels, such as WhatsApp, for communication with their peers.

**Table 5.** Channels of communication with professors and classmates during the distance study period.

| **Channels of Communication** | **With Classmates** | | **With Professors** | |
|---|---|---|---|---|
| | **Numbers** | **Percentage** | **Numbers** | **Percentage** |
| University email | 12 | 3.0 | 223 | 58.5 |
| Personal email | 14 | 3.8 | 26 | 6.7 |
| WhatsApp application | 344 | 90.3 | 131 | 34.7 |
| Other locations and apps | 11 | 2.9 | 1 | 0.1 |
| Total | 381 | 100.0 | 381 | 100.0 |

### 4.5. Student Evaluation of Previous Experience with Distance Learning Platforms

Table 6 illustrates the prior experience and training of the survey participants in relation to remote learning platforms. The majority of the students, 54.7%, did not have prior experience with remote learning, and 63.7% had not received training before the shift to remote learning (Table 6). On the other hand, 45% of the sample had prior experience with remote learning, whether it was provided by the university or obtained elsewhere.

A similar number, 36%, had received some training in working with remote or online learning platforms.

**Table 6.** Previous experiences with distance learning platforms.

| Previous Experience | Have You Pre-Tried Remote Learning Experiences during the Quarantine Period? | | Have You Had Training in Working with Remote Learning Platforms? | |
|---|---|---|---|---|
| | Numbers | Percentage | Numbers | Percentage |
| No, I have never had any previous experience | 208 | 54.7 | 243 | 63.7 |
| Yes, I had a previous experience outside the university | 91 | 23.8 | 35 | 9.3 |
| Yes, I had previous experience within the university's framework | 82 | 21.5 | 103 | 27.0 |
| Total | 381 | 100.0 | 381 | 100.0 |

Despite having previous experience with remote learning platforms, it is important to assess the ability of students to operate these platforms. The results indicate that 45.5% did not experience any major difficulties in dealing with the new technology for distance learning (Table 7). However, 36.7% of the students faced some difficulties and 17.8% faced many difficulties, that is, 54.5% of the total sample. These results highlight the lack of digital learning skills among university students and the need to develop these skills to prepare for further advanced use of any new technology in the future.

**Table 7.** Level of ability to work with distance learning platforms.

| Level of Ability | Have You Struggled with Remote Learning Platforms? | |
|---|---|---|
| | Numbers | Percentage |
| No, I didn't find any difficulty. | 63 | 16.5 |
| No, I did not find great difficulties. | 110 | 29.0 |
| Yes, I found few difficulties. | 140 | 36.7 |
| Yes, I found many difficulties. | 68 | 17.8 |
| Total | 381 | 100.0 |

*4.6. Reflection of the Emergency Distance Learning Experience on Exam Results and Level of Understanding of the Courses*

The results of the study in Table 8 showed that nearly 46% of the university students reported a higher level of understanding of the course materials during online learning under COVID-19. On the other hand, about 38% of the participants reported a decline in their comprehension of the courses, while only 15.5% of them reported no change in their level of understanding.

**Table 8.** Evaluation of the impact of distance teaching on the levels of course understanding and exam results.

| Indicator 6: The Impact of the Transition to the Distance Learning System on Academic Results. | | | | | | | | |
|---|---|---|---|---|---|---|---|---|
| Items | Indicator | Made Progress | There Is No Change | Suffered a Regression | Mean | SD | Order | Interpretation |
| The level of course understanding | n | 175 | 60 | 146 | 1.92 | 0.916 | 1 | No change |
| | % | 46 | 15.7 | 38.3 | | | | |
| The results of the examinations | n | 183 | 72 | 126 | 1.85 | 0.889 | 2 | No change |
| | % | 48.0 | 18.9 | 33.1 | | | | |
| Overall mean | | | | | 1.89 | 0.828 | | No change |

Regarding the exam results, 48% of the respondents reported an improvement in their performance, while 24.7% reported significant improvement. However, 33.1% reported a decline in their exam scores and 18.9% reported no change as a result of the shift to remote learning due to the pandemic (Table 8). The overall results show an average of 1.89 and a standard deviation of 0.828, which indicates that the students' performance was not negatively affected by the sudden change to remote learning. This demonstrates the

ability of both students and teachers to adapt to new circumstances while maintaining the effectiveness of education in non-traditional settings.

### 4.7. Impact of the Transition to Distance Learning in the Event of an Emergency on the Reshaping of the Pedagogical Communication

Table 9 combines the results of indicators 7, 8, and 9, which focus on the communication aspect of the lessons. The seventh indicator measures the continuity of communication during the pandemic, including contact with teachers and classmates. The results show an average score of 2.13 with a standard deviation of 0.854, indicating that students believe that they had continuous communication with their professors and peers. These findings highlight the importance of current modes of communication in maintaining social and educational connections during times of lockdown [23]. The eighth indicator focuses on the interactive aspect of the lessons, specifically the importance of discussions. The results indicate that students had a high level of engagement in discussions during remote learning sessions and that they continue to discuss the course material even after the lessons have ended. The overall result reveals an average score of 2.24 with a standard deviation of 0.876, showing that students felt that discussion and participation were emphasised during the remote learning sessions, and that these conversations continued on social media.

**Table 9.** Impact of the transition to distance learning on pedagogical relationships with teachers and classmates.

| Indicators | Strongly Disagree | | Disagree | | Agree | | Strongly Agree | | Mean | SD | Rank | Interpretation |
|---|---|---|---|---|---|---|---|---|---|---|---|---|
| | N | % | N | % | N | % | N | % | | | | |
| *Indicator 7* *Distance lessons during the pandemic were marked by direct and continuous contact with teachers.* | 44 | 11.5 | 64 | 16.8 | 170 | 44.6 | 103 | 27.0 | 2.13 | 0.941 | 1 | Moderate (Agree) |
| *Distance lessons during the pandemic period were marked by direct and continuous contact with classmates.* | 44 | 11.5 | 67 | 17.6 | 162 | 42.5 | 108 | 28.3 | 2.12 | 0.953 | 2 | Moderate (Agree) |
| Overall mean | | | | | | | | | 2.13 | 0.854 | | Moderate (Agree) |
| *Indicator 8* *Distance lessons during the pandemic period were characterised by intense debate and interaction.* | 68 | 17.8 | 72 | 18.9 | 140 | 36.7 | 101 | 26.5 | 2.28 | 1.045 | 1 | Moderate (Agree) |
| *Discussions about lessons continue among students on social media after the end of the session.* | 48 | 12.6 | 70 | 18.4 | 174 | 45.7 | 89 | 23.4 | 2.20 | 0.940 | 2 | Moderate (Agree) |
| Overall mean | | | | | | | | | 2.24 | 0.876 | | Moderate (Agree) |
| *Indicator 9* *After the transition to distance learning experience during the pandemic period, communication with classmates is more intensive.* | 31 | 8.1 | 52 | 13.6 | 192 | 50.4 | 106 | 27.8 | 2.02 | 2.310 | 2 | Moderate (Agree) |
| *After the transition to distance learning during the lockdown period, communication with teachers became more intensive.* | 40 | 10.6 | 114 | 29.9 | 151 | 39.6 | 76 | 19.9 | 2.31 | 0.908 | 1 | Moderate (Agree) |
| Overall mean | | | | | | | | | 2.17 | 0.801 | | Moderate (Agree) |

Note: Indicator 7: The communication dimension of lessons (continuity of communication). Indicator 8: Interactive dimension of lessons (focus of discussion). Indicator 9: Intensity of student socio-pedagogical communication after the transition to distance learning.

The ninth indicator focuses on the "Intensity of student socio-pedagogical communication after the transition to distance learning." It consists of two variables: increased communication with students after experiencing distance learning during the pandemic and increased communication with instructors after experiencing distance learning during the lockdown. The overall result shows a mean of 2.17 and a standard deviation of 0.801, indicating that students believe that the communication with their classmates and teachers became more intense after the transition. These results highlight the importance of digital tools and distance education technologies in reducing social isolation and maintaining the intensity and quality of social and educational connections.

### 4.8. Pearson Correlation

The results of the study indicate that there are four independent variables that have a significant correlation with the dependent variable, which is the intensity of student socio-pedagogical communication after the transition to distance learning. The highest positive correlation was found between the dependent variable and the interactive dimension of lessons (central discussion) (r = 766, $p < 0.001$) (Table 10). A moderate correlation was also found between the dependent variable and the communication dimension of lessons (continuity of communication) (r = 662, $p < 0.001$), the impact of moving to a distance learning system on academic outcomes (r = 501, $p < 0.001$), and the intensity of socio-pedagogical communication after the transition to distance learning. Additionally, a statistically significant negative and weak correlation ($-176$, $p < 0.001$) was found between the level of the capacity of the digital infrastructure and the dependent variable. However, the correlation coefficient was very weak; therefore, so this variable was deleted from the regression model.

**Table 10.** Pearson correlation between independent and dependent variables.

| Dependent Variable / Independent Variable | | Frequency of Use of Digital Infrastructure | The Level of Digital Infrastructure Capacity | The Impact of Moving to a Distance Learning System on Academic Outcomes | Communication Dimension of Lessons (Continuity of Communication) | Interactive Dimension of Lessons (Centrality of Discussion) |
|---|---|---|---|---|---|---|
| *The intensity of socio-pedagogical communication after the pandemic.* | **Pearson r** | 0.069 | 0.501 | 0.662 | 0.766 | −0.176 |
| | *p*-**value** | 0.180 | <0.001 | <0.001 | <0.001 | <0.001 |
| | **Total number** | 381 | 381 | 381 | 381 | 381 |

### 4.9. Multiple Linear Regression Analysis

The multiple linear regression model was used to understand the relationship between the independent variables such as the frequency of the use of digital infrastructure, the impact of the transition to distance learning on academic results, the communication dimension of lessons (continuity of communication), and the interactive dimension of lessons (centrality of the debate) and the intensity of student socio-pedagogical communication after the transition to distance learning (see Table 11).

**Table 11.** Multiple linear regression results.

| Dependent Variable | Intensity of Student Socio-pedagogical Communication after the Transition to Distance Learning | | | |
|---|---|---|---|---|
| **Model fit** | **R** | **R$^2$** | **F** | ***p*-value** |
| **Estimates** | 0.790 | 0.624 | 156.167 | 0.000 |
| **Independent variable** | **B** | **t** | ***p*-value** | **VIF** |
| **Frequency of use of digital infrastructure** | −0.030 | −0.823 | 0.411 | 1.044 |
| **The impact of moving to a distance learning system on academic outcomes** | 0.022 | 0.567 | 0.571 | 1.646 |
| **Communication dimension of lessons (continuity of communication)** | 0.234 | 5.489 | 0.000 | 2.069 |
| **Interactive dimension of lessons (centrality of discussion)** | 0.528 | 12.680 | 0.000 | 2.077 |

The multiple linear regression model showed that the combination of the independent variables explains 62.4% of the variation in the intensity of student socio-pedagogical communication after the transition to distance learning (R$^2$ = 0.624). The model (Table 11) shows that the relationship between the intensity of student socio-pedagogical communication after the transition to distance learning and the frequency of digital infrastructure usage (b = −0.030) is not statistically significant and does not fit the explanation model. Similarly, the relationship between the intensity of student socio-pedagogical communication after the transition to distance learning and the impact of switching to a remote learning system on academic performance (b = 0.022) is also not statistically significant and therefore does not fit the explanation model.

The model suggests that there is a statistically significant relationship between the communication dimension of lessons and the intensity of student socio-pedagogical communication (b = 0.234), meaning that, for every increase in the communication dimension of lessons, the density of social and pedagogical communication also improves by 0.234 units. Furthermore, the model shows that there is a statistically significant relationship between the intensity of student communication after the transition and the participatory nature of the courses (b = 0.528), indicating that an increase in the interactive nature of instruction leads to a corresponding increase in the density of social and pedagogical communication.

The multiple linear regressions table also reveals that the VIF's amplified variance factor was less than 3, indicating that there was no difficulty with multicollinearity between the model variables. The following is the regression equation:

*density of social pedagogical communication after the transition to distance learning (expected) = 0.495 + 0.234 × Dimension of communication of lessons (continuity of communication) + 0.528 × Interactive dimension of lessons (centrality of discussion) + predictive error.*

## 5. Discussion

The study found that the majority of students had access to various digital tools such as smartphones, laptops, desktops, or stationary computers during the transition to distance learning during the pandemic. The most frequently used digital tools among Ajman University students were smartphones and personal laptops. Many students had adequate access to the Internet and only a small fraction struggled with a weak and unreliable connection. This is in line with previous research that found that poor internet connection was a challenge for students during online learning and synchronous meetings [5,24–26].

The notion of distance learning that was introduced during the pandemic is called remote emergency learning/teaching [27]. This change in education methodology is a temporary and swift alteration in communication and the acquisition of skills and knowledge. The results of the analysis of the students' ability to use digital technology revealed that the majority of students had a moderate to high ability to use these tools, while a small number of students encountered significant difficulties and required assistance frequently.

Both students and teachers were required to enhance their technological proficiency to adapt to the digital classroom and learning demands brought about by COVID-19 [28]. Furthermore, the study found that most of the students used digital communication tools for more than five hours a day and spent less than five hours browsing the Internet. During distance learning, a variety of communication channels were accessible. However, within the context of formal education, most of the students communicated with their instructors through their university email, and, among peers, they utilised WhatsApp for more dynamic and responsive communication. In the context of remote emergency learning, it is crucial to have multiple communication tools to provide pedagogical and interactive channels between students and instructors, such as through class activities incorporating audio, video, text, images, emojis, and animations [29,30].

The results of the assessment of students' prior experience and training in the use of distance learning platforms indicated a general lack of digital competency among students and difficulties in the use of digital tools for remote learning. This highlights the importance of investing in the development of digital skills to prepare for future technological advancements. Previous research has emphasised the impact of digital readiness and digital skills on students' academic performance [20]. The evaluation of university students' comprehension of course material during the COVID-19 pandemic-induced distance learning period showed that less than half of the respondents had a high level of understanding, while some students made significant progress. On the contrary, more than half of the students in the sample experienced a decline or no progress in their understanding of the course material. A previous study conducted in the UAE indicated that students viewed digital learning as an effective means of improving their understanding of course material through engagement and interaction, improving their learning process through assignment feedback and providing a fun and entertaining experience [31]. These results underscore the importance of considering the implications of implementing distance or online learning in the future. Ensuring quality education through various modalities should be a priority for academic institutions, and students should have a positive learning experience and be able to effectively comprehend the material. Regarding exam results, there was an improvement for some students, although less than half experienced a decline. Previous research has shown that online exams can be more challenging if there are distractions at home and that the constant use of cameras during exams can cause anxiety [28].

The impact of the COVID-19 pandemic on the transition to distance learning has caused a reformation of socio-pedagogical communication. The results of this study showed that, on average, university students experienced an intensification of their social relationships and an increase in communication frequency for both professors and classmates, particularly during the lockdown period. Despite these advances, previous research has identified poor instructor–student communication as a major challenge faced by students during distance learning [29]. In response to this challenge, students have utilised various communication channels to interact with their instructors and classmates. For example, email communication between students and instructors was commonly used, followed by WhatsApp and social media for communication between students [32]. Thus, it is imperative for academic institutions to ensure that appropriate communication channels are in place during distance learning.

This study used a multiple linear regression model to examine the factors that influence socio-pedagogical communication after the transition to distance learning. Independent variables included the frequency of the use of digital infrastructure, the impact of remote learning on academic performance, the dimension of communication, and the participatory nature of courses. The results showed that the combination of independent variables explained 62.4% of the variation in the intensity of socio-pedagogical communication after the transition to distance learning. However, the frequency of the use of digital infrastructure and the impact of remote learning on academic performance were not found to be statistically significant in the explanation model. On the other hand, the

communication dimension of lessons and the participatory nature of courses were found to be statistically significant in the model and had a positive impact on the density of socio-pedagogical communication after the transition to distance learning.

## 6. Conclusions

This study investigated the concept of formal distance education in the context of an unexpected situation, such as the COVID-19 pandemic, where physical presence in educational and university institutions is not possible. The primary goal during such a crisis is to provide a minimum level of classes and teaching activities as quickly as possible, rather than creating a robust learning environment [30,33]. The literature highlights various theories surrounding distance learning and online learning, which are distinct from each other. Distance learning can be achieved without the use of the Internet and information and communication technology (ICT), such as messaging [34]. The focus of remote pedagogical activities is called "online teaching," while the focus on the learner is called "distance learning" or "online learning" [35,36]. As result of the responses to the COVID-19 pandemic, the transition to distance education in peripheral societies, which includes a significant portion of Arab nations, presents unique challenges. One of the most prominent difficulties is the shortage of educational and pedagogical materials available in Arabic, coupled with an increasing gap in access to information and communication technologies among different social classes. Moreover, both teachers and students have a low level of proficiency in using technology, adding to the challenges.

The findings of the study are consistent with previous research on students' experiences of online and distance learning during the COVID-19 pandemic. Although the majority of students reported having sufficient access to the Internet and proficiency in using technology, they still encountered difficulties in distance learning such as limited social interaction, inadequate communication with teachers, and an excessive reliance on social media and email. Moreover, students' access and utilisation of digital technology during the transition to distance learning during the pandemic contribute to the achievement of SDG 4 (Quality Education) and SDG 10 (Reduced Inequalities). The study found that most students had access to digital tools and adequate Internet connection; however, a small fraction struggled with unreliable internet. Therefore, investing in digital skills is crucial to prepare for future technological advancements and to ensure a positive learning experience for students. This aligns with previous findings that stress the importance of higher education institutions prioritizing the improvement of students' digital skills, particularly during the COVID-19 pandemic, by providing effective and user-friendly digital learning platforms, which can boost academic involvement [16]. The study also showed that students experienced an increase in social relationships and frequency of communication with professors and classmates. This highlights the importance of having appropriate communication channels in place for effective socio-pedagogical communication. The results of the multiple linear regression model showed that the communication dimension of the lessons and the participatory nature of the courses had a positive impact on the socio-pedagogical communication after the transition.

Although our study yielded significant findings, there is still a need for further research to deepen our understanding of the topic. The limited sample size resulting from the pandemic emergency circumstances made it challenging to collect comprehensive data, despite the encouraging results. Additionally, the study only focused on a private university in Ajman and did not include public universities that may have different results. It is also possible that the findings may differ across other countries in the Gulf region and the Middle East and North Africa, where educational infrastructure and competencies may vary. Therefore, future studies should consider increasing the sample size and expanding the scope to include a more diverse population of universities.

Despite its limitations, this study has important implications for the development of public policies related to higher education, particularly when quickly and effectively transitioning to emergency systems during crises. Our results suggest that policymakers

should prioritize the communication dimension of distance education by developing interactive content that encourages discussion and collaboration during the lessons. Moreover, students tend to use unofficial channels of communication to connect with their peers, underlining the need for inclusive communication strategies with university professors. These findings suggest that higher education experts should assess appropriate virtual communication spaces, including social media platforms and instant messaging apps, suitable for various pedagogical contexts, and that policymakers must prioritize the development of such strategies to improve their students' overall learning experience.

**Author Contributions:** Conceptualization, S.A. and S.T.; methodology, S.T. and N.A.; validation, A.S., M.E. and N.A.; data curation, S.T.; writing S.T. and S.A.; original draft preparation, S.H.; writing—review and editing, A.S. and S.T.; visualization, N.A. and S.H.; supervision, M.E.; project administration, S.A. All authors have read and agreed to the published version of the manuscript.

**Funding:** This research was funded by the Deanship of Research and Graduate Studies at Ajman University-grant number 2021-IRG-HBS-5.

**Informed Consent Statement:** Informed consent was obtained from all subjects involved in the study.

**Acknowledgments:** This article was supported by Ajman University.

**Conflicts of Interest:** The authors declare no conflict of interest.

## Appendix A

**Table A1.** The matrix of nine themes and items scales.

| Indicator | Themes | Items | Scale |
|---|---|---|---|
| 1 | The level of access to digital infrastructure | Level of access to electronic devices | High window = more than one device<br>Medium = at least one personal device<br>Weak = 1 shared device |
| | | Level of Internet access | High window = strong and stable<br>Medium = strong, unstable or stable and weak<br>Weak = unstrong and unstable |
| 2 | Frequency of use of digital infrastructure | Frequency of use of electronic devices<br>Pace of Internet surfing | High frequency = more than five hours per day<br>Medium = between 1 h and 5 h<br>Weak = less than 1 h |
| 3 | The level of employability of digital infrastructure | Level of usability of electronic devices<br>Level of access to Internet services | Easy = has no difficulty<br>Medium = has few difficulties<br>Weak = has too many difficulties |
| 4 | Pédagogie communication technique during quarantine | Channels of communication with colleagues<br>Channels of communication with teachers | Official channel = University email<br>Semi-official channel = email person<br>Unofficial channel = social media apps and sites |
| 5 | Level of access to remote learning platforms | Level of ability to work with remote learning platforms | Enable Kamel = has no difficulty<br>Medium = has few difficulties<br>Weak = has too many difficulties |
| | | Evaluate the experience of working with remote learning platforms | Positive<br>Moderate<br>Negative |
| 6 | The impact of moving to a distance learning system on academic outcomes | The impact of moving to a distance learning system on understanding<br>The impact of moving to distance learning on exam results | Progress made<br>No change<br>He suffered a retreat |
| 7 | The Communication dimension of lessons (continuity of Communication) | Lessons during the pandemic period were marked by direct and continuous contact with teachers<br>Distance lessons during the pandemic period were marked by direct and continuous contact with colleagues and colleagues | Strongly agree<br>Agree<br>Disagree<br>Strongly disagree |
| 8 | Interactive dimension of lessons (centrality of discussion) | Distance lessons during the pandemic period were characterised by a lot of discussion and interaction between students and professors<br>Discussions about lessons continue among colleagues on social media platforms after the end of the lesson | Strongly agree<br>Agree<br>Disagree<br>Strongly disagree |

**Table A1.** *Cont.*

| Indicator | Themes | Items | Scale |
|---|---|---|---|
| 9 | The intensity of student social-pedagogical communication after the transition to distance learning | After the transition to distance learning experience during the pandemic period, communication with classmates is more intensive<br>After the transition to distance learning during the lockdown period, communication with teachers became more intensive | Strongly agree<br>Agree<br>Disagree<br>Strongly disagree |

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
