# Peer review of "Transition to Distance Learning: Student Experience and Communication during the COVID-19 Pandemic in the United Arab Emirates"

_sustainability, doi:10.3390/su15086456_

Round 1

Reviewer 1 Report

Dear Authors,

Thank you for submitting this manuscript. Although the topic is interesting, the scope of the article, both in terms of size of the sample and geographical focus, is too narrow to allow a generalizability of findings at international level. To give scientific rigor to the paper, I suggest to authors:

·       To extend the literature review;

·       To increase the number of respondents or to change the methodology of the work;

·       To submit the paper to journals specifically devoted to education sciences.

Best wishes!

Author Response

Dear Reviewer, 

Thank you very much for your valuable feedback and comments. We carefully adjusted the paper, including your comments and suggestions, which helped us to improve the manuscript. We updated our literature review, added the recommended literature as suggested, and added new references across the text.

Please find attached the detailed responses to each of your comments.

Reviewer 2 Report

An article made with a very good arrangement, the author should make further studies of the topic because the pandemic is over. and whether there have been any changes or developments in the implementation of post-pandemic learning.

Author Response

(The authors gave the same response as above.)

Reviewer 3 Report

Thanks to the editors for the opportunity to comment on the submitted article: “ Transition to distance learning: Student experience and communication during COVID-19 pandemic in the UAE.“ I recommend not using the abbreviation in the title of the article „UAE“ (United Arab Emirates).

I find the paper interesting and well-written. The paper is well-organizing and exposes clearly the research developed. However, I have a question/suggestion related to the abstract. In the abstract, I recommend clearly formulating the goal and methods which are presented in the text.

I recommend emphasizing more in the text what makes the study innovative and why it is/was necessary to solve this problem. It would also be appropriate to highlight for whom the results of the study are useful and for whom the results are a help - the main implementers of the results.

The authors work with appropriate literary sources (but can be extended). The submitted text can be considered as a professional study and I recommend it for publication after modifications.

Author Response

(The authors gave the same response as above.)

Reviewer 4 Report

The article presents the current issue related to the need to use remote teaching technicians during the COVID-19 pandemic. The article reads well, however, I have three remarks:

1. How was the random sampling of students carried out? Was it stratified (by gender, year of study, age, place of residence) – therefore, was the sample representative? Can the results be extrapolated to the entire student community at the researched university?

2. The research results could be more interesting if they were also presented through the prism of, for example, gender or age group. I assume that the respondents included this data in the form - the article gives the structure of respondents in these dimensions. Maybe there would be statistically significant relationships (e.g. using the Chi-square method)?

3. The literature review was based on a relatively small number of references. Expanding sources to at least 50 is recommended

Examples of references that could be added:

Ochnio, L.; Rokicki, T.; Czech, K.; Koszela, G.; Hamulczuk, M.; Perkowska, A. Were the Higher Education Institutions Prepared for the Challenge of Online Learning? Students’ Satisfaction Survey in the Aftermath of the COVID-19 Pandemic Outbreak. Sustainability 2022, 14, 11813. https://doi.org/10.3390/su141911813

Paliszkiewicz Joanna, SkarzyÅ„ska Edyta, In: Online Learning Analytics / Liebowitz Jay (ed.), Data Analytics Applications, 2022, New York, Taylor & Francis Group, s.197-212, ISBN 9781000538939

ParliÅ„ska, A. (2022). E-LEARNING CHALLENGES IN POLAND DURING THE COVID-19 PANDEMIC. Zeszyty Naukowe SGGW, Polityki Europejskie, Finanse I Marketing, (27(76), 80–88. https://doi.org/10.22630/PEFIM.2022.27.76.7

I believe that taking into account the above comments will increase the substantive value of the article and increase the reach of its future readers.

Author Response

Dear Reviewer, 

Thank you very much for your valuable feedback and comments. We carefully adjusted the paper including your comments and suggestions which helped us to improve the manuscript. We updated our literature review, added the recommended papers as it was suggested and added new references across the text.

Please find attached the detailed responses to each of your comments.

Round 2

Reviewer 1 Report

Dear Author(s),

Thank you for revising the manuscript, which has now consistently improved with respect to the previous version. Before final acceptance, please try to extend conclusions including considerations about the limitations of your work and its possible possible policy/practical implications.

Best wishes!

Author Response

Dear Reviewer, 

Thank you for providing your valuable feedback and suggestions. We have revised the paper based on your input and have expanded the conclusion section. To make it easier to identify the changes, we have highlighted the added paragraphs in yellow within the conclusion section of the revised paper.

We appreciate your time and effort in reviewing our work and providing us with constructive criticism. Your comments have been invaluable in helping us improve the quality of our research. Thank you again for your contribution to our study.
